# In Situ Electrochemical Synthesis of Squamous-like Cu_2_S Induced by Sulfate-Reducing Bacteria as a Fenton-like Catalyst in Wastewater Treatment: Catalytic Performance and Mechanism

**DOI:** 10.3390/nano14070621

**Published:** 2024-04-02

**Authors:** Liuhui Zhao, Zihao Li, Jing Yang, Jiawen Sun, Xiaofan Zhai, Fubin Ma, Jizhou Duan, Peng Ju, Baorong Hou

**Affiliations:** 1School of Biologic Engineering, Qilu University of Technology (Shandong Academy of Sciences), Jinan 250300, China; 10431210980@stu.qlu.edu.cn (L.Z.);; 2CAS Key Laboratory of Marine Environmental Corrosion and Bio-Fouling, Institute of Oceanology, Chinese Academy of Sciences, No. 7 Nanhai Road, Qingdao 266071, China; 3Sanya Institute of Oceanology, South China Sea Institute of Oceanology, Chinese Academy of Sciences, Zhenzhou Road, Sanya 572000, China; 4Guangxi Key Laboratory of Marine Environmental Science, Institute of Marine Corrosion Protection, Guangxi Academy of Sciences, Nanning 530007, China; 5Key Laboratory of Marine Eco-Environmental Science and Technology, Marine Bioresource and Environment Research Center, First Institute of Oceanography, Ministry of Natural Resources, No. 6 Xianxialing Road, Qingdao 266061, China

**Keywords:** squamous-like Cu_2_S, SRB, wastewater treatment, electrolysis

## Abstract

In this paper, a novel method was proposed for the synthesis of Cu_2_S on copper mesh via electrolysis in SRB culture medium. It was found that following electrolysis in SRB medium, squamous-like Cu_2_S arrays were obtained on the copper mesh, and the Cu_2_S loading contents varied with the electrolyzing parameters. The resultant Cu_2_S on copper mesh in SRB (CSCM-SRB) with the highest catalytic MB degradation properties was produced by electrolysis at 3.75 mA/cm^2^ for 900 s. The optimized MB-degrading conditions were determined to be 1.2 cm^2^/mL CSCM-SRB with 0.05 M H_2_O_2_ at 35 °C when pH = 6, under which the degradation of MB reached over 99% after 120 min of reaction. Disinfecting properties was also proven by antibacterial tests, revealing that an almost 100% antibacterial rate against *E. coli* was obtained after 8 min. The organic compounds produced by SRB adsorbed on CSCM-SRB strongly promoted the degradation of MB. Furthermore, possible Fenton-like mechanisms of CSCM-SRB were proposed, illustrating that ·O_2_^−^, ·OH, and ^1^O_2_ acted as the main functional species during Fenton-like reactions, leading to effective MB degradation and high antibacterial properties. Finally, a simple device for wastewater treatment was designed, providing possible applications in real environments.

## 1. Introduction

In recent decades, due to rapid industrial development, water contamination has attracted much attention. Many kinds of pollutants, such as organic dyes and harmful microorganisms, have been detected in ground, sewage, and drinking water resources, leading to increased risks of human disease and mortality [1,2]. Therefore, advanced and green materials for wastewater treatment with low costs and high efficiency are still urgently required. Thus far, several methods have been applied to control water contamination, including physical methods (adsorption, ion exchange, liquid–liquid extraction, etc.) [3,4], chemical methods (chemical oxidation, photocatalytic degradation, etc.) [5,6], and biological methods (activated sludge method, etc.) [7,8,9]. However, traditional techniques have shown low reaction rates, high applying concentrations, and high toxicity. In addition, common industrial wastewater always contains abundant and complex organic matter, making it difficult for it to be completely decontaminated [10]. Thus, advanced oxidation methods have been proposed for novel water contamination due to their high efficiency and nontoxicity [11].

The Fenton reaction, as one of the most popular advanced oxidation methods, can produce hydroxyl radicals and superoxide radicals as the main active oxidizing substances for organic pollutant degradation and bacterial disinfection. The Fenton reaction was initially proposed by H. J. H. Fenton in 1894 [12]. In the Fenton reaction, H_2_O_2_ reacts with Fe(II) to generate OH· with a high oxidizing property whose oxidizing potential (pH = 4) can reach 2.73 V [13]. Thus, a degradation process based on the Fenton reaction can deal with aromatic compounds and heterocyclic compounds which are difficult for traditional methods. Some researchers have found that other excessive metal ions, such as Co(II), Cr(III), Cu(II), and Cu(I), can also catalyze H_2_O_2_ to produce active substances and degrade organic pollutants. These are named Fenton-like reactions [14]. Fenton-like reactions are considered highly promising for pollutant degradation and bacterial disinfection. Among Fenton-like catalysts, Cu_2_S has attracted much attention due to its low costs, thermal stability, and insolubility in both water and organic solvents.

However, several issues in the application of Cu_2_S in wastewater treatment remain unsolved. Firstly, the Cu_2_S chemical synthesis process is complicated and has strict synthesizing conditions. Currently, although Cu_2_S nanomaterials with various structures such as nanowires, nanosheets [15], nanorings [16], spherical nanoflowers [17,18], and hollow nanospheres [19] have been successfully prepared, the synthesizing methods mainly depend on gas-phase and liquid-phase methods. For example, Zhanguo Su et al. used the hydrothermal method to calcinate thiourea and copper nitrate in a reactor at 230 °C for 72 h and then sintered them at 700 °C in a vacuum environment for 6 h to obtain wormlike Cu_2_S [20]. Siyuan Luo et al. used the atmosphere method to introduce a mixture of pure H_2_S and O_2_ into a sealed gas chamber containing copper foil and reacted them for more than 10 h to obtain a fluffy Cu_2_S film on the surface of the copper foil [21]. Liu et al. synthesized ultra-thin Cu_2_S nanowires via the hydrothermal method using CuCl_2_ and NaS_2_CNEt_2_ as auxiliary surfactants and reacting them at 160 °C for 12 h [22]. Thus, the existing techniques used to synthesize Cu_2_S have involved complicated reacting steps and strict synthesis conditions with relatively long reaction times.

Secondly, the catalytic efficiency and biocompatibility of Cu_2_S still need to be improved for real applications. Peng Meng et al. investigated the pollutant degradation efficiency of Cu_2_S, finding that both linear and flower-shaped Cu_2_S showed poor photocatalytic degradation effects on organic pollutants (X-3B) (11.7–50%). However, in the presence of Fe(II) and EDTA, the Fenton degradation of Cu_2_S against X-3B was dramatically improved [23]. To improve the properties of Cu_2_S, many attempts have been made to composite Cu_2_S with other catalysts. For example, Sharma, Saurabh K. et al. obtained Cu_2_S/SnO_2_ nanocomposites using the precipitation method. The Cu_2_S/SnO_2_ nanocomposites showed a 98% degradation efficiency against MB after 210 min of photocatalytic reaction [24]. Xu Feng et al. also synthesized Cu_2_S/ZnO composite films on conductive glass, revealing a degradation rate of 60–86% against methylene orange [25]. In addition, chemically synthesized Cu_2_S would cause rejection reactions in organisms, indicating potential environmental problems [26].

Facing these two problems, bio-synthesized Cu_2_S may act as an ideal solution for the application of Cu_2_S in wastewater treatment, as the biosynthesis process is relatively environmentally friendly and low-energy-consuming and the biosynthesized materials are highly biocompatible. Thus, the sulfur source for Cu_2_S can be derived from sulfur-functional bacteria, such as sulfate-reducing bacteria (SRB) [27]. SRB are widely distributed in marine sediments, oil fields, and even the human gut [28,29]. SRB have been reported to be effective functional bacteria for biosynthesized metal sulfides, such as ZnS, NiS, FeS, Cu_2_S, MnS, etc. [27,30]. SRB utilize SO_4_^2−^ as the terminal electron acceptor, which is converted to several sulfide forms, such as H_2_S, HS^−^, and S^2−^. These sulfide ions can act as the sulfur source for Cu_2_S [31]. A similar synthesis strategy was also proposed by Yang Huan et al. in 2020. Ni(Fe)OOH-FeS was biosynthesized by SRB to obtain high oxygen evolution properties in alkaline electrolytes, revealing a current density of 10 mA/cm^2^ under an overpotential of 220 mV [32].

Although the properties of biosynthesized materials have been proven to be highly improved, the mechanism of action of SRB-induced catalytic materials remains unknown. This is because SRB biosynthesis includes both precipitation reactions and microbial mineralization processes. The extracellular polymers (EPS) produced by SRB would greatly influence the nucleation and growth of minerals due to their negative charge and their organic three-dimensional framework composed of proteins, polysaccharides, and humic acids, which could participate in the biosynthesis process [33,34]. These factors highly influence the biological and catalytic properties of the resultant materials, which should be discussed in detail for biosynthesis research.

In this work, a copper mesh was used as the substrate for Cu_2_S synthesis induced by SRB. The electrochemical method was also used during the biosynthesis process to control Cu(I) generation. Thus, a novel squamous-like Cu_2_S was deposited in situ on the copper mesh. The morphology and structure of the biosynthesized Cu_2_S were studied and excellent Fenton-like catalytic performance was found. The application of the biosynthesized Cu_2_S in organic dye degradation and sterilization was further researched. Moreover, a conceptual device based on the biosynthesized Cu_2_S deposited on the mesh was constructed, revealing its prospective application in wastewater treatment.

## 2. Experimental

### 2.1. Culture of SRB

The bacteria strain used in this study was identified as *Desulfovibrio bizertensis* SY-1 and was isolated from the South China Sea. Postgate C medium (PGC) was employed for the SRB culture and was prepared with 0.5 g KH_2_PO_3_, 1 g NH_4_Cl, 0.06 g CaCl_2_·6H_2_O, 6 mL 70% sodium lactate solution, 1 g yeast extract, 0.3 g sodium citrate, and 0.06 g MgSO_4_ dissolved in 1 L of filtered seawater. Then, the pH of the PGC was adjusted to 7.2 and divided into 500 mL anaerobic bottles, in which flowing nitrogen was continuously infused for 40 min for deoxygenation. After sterilization at 121 °C and 101 MPa for 20 min, the PGC medium was inoculated by 1% *v*/*v* SRB 5-day-old SY-1 planktonic culture and then cultured for 8 days. It was used for subsequent material biosynthesis [35].

### 2.2. Synthesis of Cu_2_S on Copper Mesh Induced by SRB

The copper mesh used in this experiment was purchased from Churui metals company, Hebei. The copper mesh, with a size 20 mm × 30 mm × 0.2 mm, was ultrasonically cleaned in ethanol for 15 min before use. Before use, the copper mesh was pretreated by ultrasound in 3 mmol L^−1^ HCl for 2 min to remove the oxide on the copper mesh previously formed in air and then adequately washed with deionized water. Then, the pretreated copper mesh was used as the working electrode to assemble an electrolytic cell, as shown in Figure 1. In this cell, a platinum electrode was employed as the counter electrode and a saturated calomel electrode (SCE) was used as the reference electrode. The SRB culture medium described in Section 2.1, acting as the electrolyte, was transferred into the electrolytic cell under anaerobic conditions, and an electrochemical workstation (CHI 760, Chenhua, Shanghai, China) was employed to provide stable current for electrolysis.

During electrolysis, several parameters were changed in the process, including gradient current density (0.25 mA/cm^2^, 1.25 mA/cm^2^, 2.5 mA/cm^2^, 3.75 mA/cm^2^, and 5 mA/cm^2^, respectively, with an electrolysis time of 900 s) and electrolysis times (300 s, 600 s, 900 s, 1200 s, and 1500 s, respectively; the optimal current density was used in this case). Agitation at 100 rpm was also performed along with the electrolysis to ensure the homogeneity of the electrolyte. After electrolysis, the surface of the copper mesh turned uniformly black. The resultant coupon was named CSCM-SRB.

### 2.3. Characterization of CSCM

The morphology of Cu_2_S on the surface of the copper mesh was observed using scanning electron microscopy (SEM, ULTRA 55, Zeiss, Oberkochen, Germany) and transmission electron microscopy (TEM, JEM-2100F, JEOL, Tokyo, Japan). Additionally, the materials formed on the surface of the copper mesh were qualitatively analyzed using X-ray energy dispersive spectroscopy (EDS, Detector Model 550i, IXRF, Austin, TX, USA), X-ray diffraction analysis (XRD, Max–3C, Rigaku D, Tokyo, Japan), and X-ray photoelectron spectroscopy (XPS, ESCALAB 250XI, Thermo Fisher, Waltham, MA, USA).

### 2.4. Fenton-like Degradation of Methylene Blue (MB) by CSCM-SRB

After electrolysis, a black CSCM-SRB with a size of 20 mm × 20 mm was obtained and cut to a specific area. CSCM-SRB and H_2_O_2_ were employed to degrade 10 mg/L MB. During the 120 min reaction, the absorbance of the degraded MB solution at 665 nm was measured by a UV-Vis DRS spectrophotometer for 30 min [36]. CSCM-SRB usage was set as 0, 0.6, 0.8, 1.0, 1.2, 1.4, and 1.6 cm^2^/mL. The concentration of H_2_O_2_ was set as 0, 0.03, 0.04, 0.05, 0.06, 0.07, 0.08, and 0.09 M. pH and temperature were also optimized. The pH was adjusted with NaOH using a pH meter, and the temperature was controlled by a thermostat water bath. Except the variate, other parameters were set as follows: H_2_O_2_ concentration 0.05 M, CSCM-SRB usage 0.8 cm^2^/mL, pH 6, and temperature 25 °C.

### 2.5. Disinfection Properties of CSCM-SRB against E. coli

Luria–Bertan medium (LB medium) consisting of 10 g L^−1^ NaCl, 10 g L^−1^ tryptone, and 5 g L^−1^ yeast extract was prepared and sterilized at 121 °C for 30 min for *E. coli* culture. 1 *v/v*% 12 h old *E. coli* was inoculated into the LB medium and cultured at 35 °C for 10 h. Then, the bacterial bodies were separated by high-speed centrifuge (4500 rpm, 3 min) and then suspended in phosphate-buffered saline (PBS, 8.0 g NaCl, 0.2 g KCl, 1.44 g Na_2_HPO_4_, and 0.24 g KH_2_PO_4_ dissolved in 1 L deionized water) to obtain a 10^8^ cfu/mL *E. coli* PBS suspension. Then, four groups of disinfection experiments were conducted in the obtained *E. coli* PBS suspension: (1) CSCM-SRB (0.8 cm^2^/mL) together with H_2_O_2_ (0.05 M); (2) H_2_O_2_ (0.05 M); (3) CSCM-SRB (0.8 cm^2^/mL); and (4) untreated copper mesh (0.8 cm^2^/mL). After reacting for 0, 2, 4, 6, and 8 min, the residual concentration of living bacteria was determined by the CFU method [37]. Moreover, SEM observation of the *E. coli* PBS suspension and fluorescence microscope observation of CSCM-SRB were conducted.

### 2.6. Mechanism Study on CSCM-SRB

To clarify the effect of SRB on CSCM-SRB synthesis, attempts were made to synthesize CSCM in abiotic electrolytes. To unify the chemical factors, S^2−^ concentration of the SRB electrolyte was determined by inductively coupled plasma atomic emission spectrometry (ICP-AES, Agilent 720 ES, Shanghai, China). Then, equivalent Na_2_S was added into the PGC medium (pH = 7.2) and NaCl solution (pH = 7.2), respectively, to obtain two abiotic electrolytes. Under the same electrolysis conditions, CSCM-Na_2_S(PGC) and CSCM-Na_2_S(NaCl) were obtained and analyzed by XRD.

To clarify the effect of SRB on CSCM-SRB properties, an elution experiment was performed. To remove the biogenic imprint, the prepared CMSC-SRB was immersed in 1% sodium dodecyl sulfate (SDS) detergent and heated for 30 min at 95 °C, rinsed with a large amount of deionized water, and dried with nitrogen, thus producing CMSC-el [38]. Fourier-transform infrared spectroscopy (FT-IR, vertex 70, Bruker, Karlsruhe, Germany) was employed to analyze the organic components of CMSC-SRB and CMSC-el. Furthermore, the MB degradation test as described in Section 2.4 was performed.

Finally, the holes and free radicals generated by CMSC-SRB and CMSC-el were determined using Electron Paramagnetic Resonance (EPR, EMXplus, Bruker, Karlsruhe, Germany). A free radical-trapping experiment was employed by adding the corresponding scavengers into the catalytic procedures. Tert-Butanol (TBA), p-benzoquinone (BQ), and Furfuryl alcohol (FFA) were used as the scavengers of ·OH, ·O_2_^−^, and ^1^O_2_, respectively [39].

## 3. Results and Discussion

### 3.1. Characterization of CSCM-SRB

After electrolysis synthesis, the obtained CSCM-SRB groups were analyzed systematically to clarify their chemical components. As shown in Figure 2a,b, the diffraction peaks of Cu_2_S as well as Cu appeared on CSCM-SRB. The crystalline orientations of Cu_2_S corresponded with the orthorhombic system according to JCPDS No. 83-1462. With increased current density and electrolysis time, sharp crystal diffraction peaks were enhanced, showing that orthorhombic Cu_2_S crystals had formed on the copper mesh under high current density and long electrolysis times. EDS analysis was further performed to elucidate the elemental composition of the resultant CSCM-SRB. Only Cu was found on the copper mesh, while both Cu and S were found on CSCM-SRB (Figure 2c), indicating the formation of Cu_2_S. In addition, the EDS map shown in Figure 2d–f showed that Cu and S were distributed uniformly on the prepared CSCM-SRB. Further analysis was performed focusing on the S contents of CSCM-SRB electrolyzed under different conditions based on the EDS results. As Figure 2c shows, S content increased significantly as current density went up, and peak S content was obtained when current density was set to 3.75 mA/cm^2^. The variation in S content vs. electrolysis time showed similar tendencies, and peak S content was obtained at an electrolysis time of 900 s. Thus, the highest loading content of Cu_2_S was obtained at the optimized condition of 3.75 mA/cm^2^ and 900 s.

The morphology of CSCM-SRB was observed by SEM and TEM. Figure 3a–c shows the morphology of the untreated copper mesh, and smooth surfaces were found on the substrate. On CSCM-SRB synthesized under 3.75 mA/cm^2^ for 900 s, plenty of nanocubes were found on the substrate, forming a squamous-like structure (Figure 3d–f) These nanocubes showed a compact and random arrangement with a size of ~1 μm. Additionally, CSCM-SRB synthesized under other electrolysis conditions showed similar characteristics (see Appendix A). As electrolysis time was extended, the grain size of Cu_2_S firstly increased but then remained similar after 900 s (Appendix A), which might suggest a dynamic balance for this surface modification. Moreover, as current density went up, the Cu_2_S nanocubes showed an increased grain size but a smooth arrangement (Appendix A). It is known that the nucleation rate of crystals has been found to be positively correlated with overpotential and with current density [40]. Here, on CSCM-SRB, increased nucleation driving forces for the Cu_2_S nanocubes were caused by a relatively high current density, leading to large grain sizes for coarse crystals or dendrites. Therefore, an optimized current density and electrolysis time lead to compact and uniform squamous-like Cu_2_S films. TEM observations were further performed and are shown in Figure 3g,h. Squamous Cu_2_S corresponding to the SEM image could be clearly distinguished, and a lattice spacing of ~0.22 nm appeared, which was indexed to the (103) crystal plane of Cu_2_S and was also verified by selected area electron diffraction (SAED) characterization (Figure 3i).

To further understand the composition and valence states of Cu and S in Cu_2_S on the surface of CSCM-SRB, XPS analysis was performed, and the results are presented in Figure 4. The peaks of Cu 2p, O 1s, C 1s, and S 2p were observed on the survey spectrum, revealing that Cu, S, O, and C elements existed on the CSCM-SRB surface. Cu and S were assigned to electrolyzed copper sulfide, while C and O came from the protein adsorbed on the CSCM-SRB during the bacterial medium electrolysis. In Figure 4b, the peaks of Cu in high-resolution spectra appearing at a binding energy of 932.5 eV and 952.3 eV correspond to Cu^+^ 2p3/2 and Cu^+^ 2p1/2, respectively [20]. In addition, a satellite peak at ~943 eV was also found, illustrating the existence of Cu(I) on CSCM-SRB. In the case of S 2p (Figure 4c), the peaks appearing at 160.5–164.5 eV could be divided into three peaks: the peak at 160.8 eV representing metal sulfides and the peaks at 163.2 eV and 162.1 eV being attributed to S^2−^ p1/2 and S^2−^ p3/2 [15]. Thus, the presence of S(-II) on CSCM-SRB is confirmed, illustrating the stable existence of Cu_2_S on CSCM-SRB.

Summarizing the data obtained by comprehensive characterizations, squamous-like Cu_2_S arrays with a size of ~1 μm were obtained on copper mesh, exhibiting typical orthorhombic characteristics. The highest Cu_2_S loading contents were synthesized under 3.75 mA/cm^2^ for 900 s.

### 3.2. Fenton-like Degradation of MB by CSCM-SRB

To evaluate the Fenton-like catalytic properties of CSCM-SRB, an MB degradation experiment was performed and is shown in Figure 5. Following a 120 min reaction, the UV-Vis DRS spectrum of the system (0.8 cm^2^/mL CSCM-SRB synthesized by 2.5 mA/cm^2^ for 900 s with 0.05 M H_2_O_2_ at pH = 6 at 25 °C) was measured for 30 min (Figure 5a). The absorption peak of MB appears at 665 nm, representing the concentration of MB (*C*_0_). After 120 min of reaction, the peak at 665 nm is strongly weakened, showing that the concentration of MB (*C*_t_) is obviously reduced. Meanwhile, the blue color of the MB solution turned transparent (colorless) after 120 min of reaction. Then, degradation efficiency was calculated by *C*_t_/*C*_0_ to optimize the electrolysis and reacting parameters and is shown in Figure 5b,c,f–i.

The optimized current density applied for CSCM-SRB electrolysis was firstly determined via degradation efficiency (degradation conditions of 0.8 cm^2^/mL CSCM-SRB and 0.05 M H_2_O_2,_ at pH = 6 at 25 °C). In Figure 5b, we see that when no current was applied, the degradation effect was relatively slight due to the slow dissolution of Cu^+^ from the copper mesh in the SRB medium to form infinitesimal Cu_2_S with Fenton-like properties. As the current density increased from 0.25 to 5.00 mA/cm^2^, the degradation efficiency firstly increased and then decreased. The highest degradation efficiency was ~75% and was obtained after 120 min at 3.75 mA/cm^2^. 

Electrolysis time was further studied from 300 to 1500 s under an optimized current density of 3.75 mA/cm^2^ (Figure 5c). Various times of the resultant CSCM-SRBs electrolysis were tested under 0.05 M H_2_O_2_ pH = 6 and 0.8 cm^2^/mL CSCM-SRB at 25 °C. Electrolysis times did not show a significant influence on the degradation efficiency of MB. In general, long electrolysis times led to high degradation efficiency. An electrolysis time of 900 s resulted in the highest degradation efficiency of ~75% after 120 min and was considered the optimized electrolysis time. Thus, a current density of 3.75 mA/cm^2^ and an electrolysis time of 900 s were determined to be the optimized electrolysis parameters, which was consistent with the SEM and EDS results. Under this condition, the resultant CSCM-SRB showed the highest Cu_2_S content with the most compact morphologies. Kinetic analyses of MB degradation catalyzed by CSCM-SRB with various electrolysis times were performed and the results are shown in Figure 5d. The kinetics of the degradation reaction was fitted to a pseudo-first order reaction [41] as follows: ln*C*_0_*/C*_t_ = *kt* + a, where *k* represents the reaction rate; *C*_0_ represents the initial concentration of MB; and *C*_t_ represents the concentration of MB after reacting for time *t*. The highest *k* value was determined to be 0.0224 h^−1^ at a current density of 3.75 mA/cm^2^ and electrolysis time of 900 s. 

The reacting conditions, including H_2_O_2_ concentration, pH value, reacting temperature, and CSCM-SRB usage, were further optimized as shown in Figure 5e–h. In Figure 5e, MB degradation efficiency vs. reacting time are compared under various H_2_O_2_ concentrations with 0.8 cm^2^/mL CSCM-SRB at pH = 6 at 25 °C. When no H_2_O_2_ was added, CSCM-SRB would adsorb about 40% MB during the initial 30 min, leading to an obvious decrease in *C*_t_/*C*_0_ value. When some H_2_O_2_ was added, the degradation efficiency continued to increase for 30~120 min, and the highest degradation efficiency of ~85% was obtained after 120 min at an H_2_O_2_ concentration of 0.05 M.

Subsequently, pH was then optimized with 0.05 M H_2_O_2_ and 0.8 cm^2^/mL CSCM-SRB at 25 °C, as shown in Figure 5f. It was found that when pH = 4–8, the degradation efficiency remained stable and over 80%, and the highest degradation performance occurred at pH = 6. However, when the reacting system had a strong acidic (pH = 3) or strong alkaline (pH = 9) pH value, its degradation performance was strongly weakened. It has been reported that the Fenton reaction always shows good properties under acidic conditions and that about pH = 4 is normally best, which greatly limits its application in real environments [42]. In this case, CSCM-SRB proved to be effective in a relatively large pH range, which illustrates its high stability. This characteristic could be attributed to the bacterial participation in the synthesis process of CSCM-SRB. During electrolysis, macromolecular organic substances secreted by the SRB, such as proteins and EPSs, absorb and participate in the synthesis of Cu_2_S, leading to a Cu_2_S–protein complex with low sensibility towards pH.

CSCM-SRB usage was considered another important parameter for MB degradation. In Figure 5g, the use of 0.6–1.6 cm^2^/mL CSCM-SRB was studied under 0.05 M H_2_O_2_ at pH = 6 at 25 °C. As CSCM-SRB usage increased, degradation efficiency obviously grew. When CSCM-SRB usage went up to 1.2 cm^2^/mL, degradation efficiency remained stable at ~90%. Considering resource-saving and commercial aspects, 1.2 cm^2^/mL was chosen as the optimized CSCM-SRB usage.

Finally, temperature was also found to greatly contribute to the degradation effect. Temperature optimization was performed with 0.05 M H_2_O_2_ and 1.2 cm^2^/mL CSCM-SRB at pH = 6 and the results are shown in Figure 5h. As the temperature went up, the degradation efficiency was first promoted and then weakened. The highest degradation performance was achieved at 35 °C, with degradation efficiencies of 98% (60 min) and over 99% (120 min). These relatively mild reacting conditions make CMSC-SRB suitable for real wastewater treatment. Furthermore, when kinetic analysis was performed, the highest *k* value was determined to be 0.0574 h^−1^ at a reacting temperature of 35 °C.

To sum up, the CSCM-SRB with the highest catalytic degradation properties was prepared by electrolysis at 3.75 mA/cm^2^ for 900 s. The optimized reacting conditions were determined to be 1.2 cm^2^/mL CSCM-SRB with 0.05 M H_2_O_2_ at pH = 6 at 35 °C. Under this condition, the degradation of MB reached over 99% after 120 min of reaction. Thus, subsequent studies were performed using CSCM-SRB electrolyzed at 3.75 mA/cm^2^ for 900 s under these optimized parameters.

### 3.3. Fenton-like Disinfection Properties of CSCM-SRB

In wastewater treatment, bacteria and organic dyes are considered to be the main pollutants. Thus, subsequently, the Fenton-like disinfection properties of CSCM-SRB were studied in 10^8^ cfu/mL *E. coil* PBS suspension under the optimized conditions described above.

As shown in Figure 6a, the systems containing either only copper mesh or CSCM-SRB showed no obvious influence on the concentration of the bacteria, while the system with only H_2_O_2_ showed a slight inhibition effect on *E. coli*. However, the system with both CSCM-SRB and H_2_O_2_ showed excellent antibacterial effects. After 4 min reaction, the bacteria could not be detected by the CFU method, revealing an almost 100% antibacterial effect. Then, the bacteria attached on the mesh were evaluated by fluorescence microscopy observation (Figure 6b,c). As the images show, relatively weak green fluorescence appeared on CSCM-SRB with H_2_O_2_ (Figure 6b), illustrating low attachment on CSCM-SRB, while on CSCM-SRB without H_2_O_2_ (Figure 6b), strong green fluorescence was found on the mesh, showing high bacterial attachment. These results also corresponded with the results of bacterial concentration in the medium. Then, the pelagic bacterial body in the PBS suspension was further observed by SEM. Before the reaction, the *E. coli* bacteria showed bacilliform cells with smooth surfaces, indicating typical living bacteria. After a 4 min reaction in the system with both CSCM-SRB and H_2_O_2_, the membrane of the bacteria was damaged and the cellular contents were leaking. Then, after an 8 min reaction, the bacterial body shriveled to pieces, and no living bacteria were found in the medium. These observations show that the reaction strongly attacked the bacterial membrane, which is typical of an oxidation-type bactericide, leading to excellent antibacterial properties.

### 3.4. Study on the Fenton-like Mechanism of CSCM-SRB

It has usually been believed that the formation of Cu_2_S is precipitated by electrolyzed Cu^+^ and SRB-produced S^2-^. However, in this study, an interesting finding was obtained in that Cu_2_S was obtained in neither PGC + Na_2_S electrolyte nor NaCl + Na_2_S electrolyte under the same electrolysis conditions (Figure 7a). Moreover, PGC + Na_2_S electrolyte and NaCl + Na_2_S electrolyte contained the same concentration of S^2−^ with the cultured SRB medium, as determined by ICP-AES. No diffraction peaks of Cu_2_S were found for CSCM-Na_2_S(PGC) and CSCM-Na_2_S(NaCl), while obvious diffraction peaks of Cu_2_S appeared for CSCM-SRB, illustrating not only the participation of Cu^+^ and S^2−^ in precipitation, but also the significant contribution of organic substances produced by bacteria, such as proteins, to the formation of Cu_2_S. This could be attributed to the fact that the positive/negative charge of the proteins produced by SRB would attract Cu^+^ and S^2−^, thus promoting the agglomeration and crystallization of Cu_2_S.

Further investigations were conducted on eluted CSCM-SRB, named CSCM-el, to remove the proteins which might attach to CSCM surfaces. A FT-IR spectrogram of CSCM-SRB and CSCM-el was performed and is shown Figure 7b. For CSCM-SRB, the strong absorption peaks around 3350 cm^−1^ could be attributed to O-H and N-H stretching vibrations, while the peaks at 2350 cm^−1^ represent the triple bonds in biomolecules. Also, the peak at 1633 cm^−1^ is considered to be associated with organic C chains [38]. These characteristic peaks show that abundant organic compounds, probably proteins, are absorbed on the surface of CSCM-SRB. Moreover, no obvious peaks were found on CSCM-el; only some peaks under 800 cm^−1^ originating from the metal sulfide Cu_2_S were found [43], indicating that the biogenic organic compounds, mainly proteins, were removed from the surface of CSCM-SRB. Thus, subsequently, the degradation efficiency of MB was further evaluated, as shown in Figure 7c. It was found that no obvious adsorption occurred on CSCM-el, and the final degradation efficiency was less than 40%, while CSCM-SRB with proteins reached a degradation efficiency of over 90%. The removal of the proteins absorbed on CSCM dramatically weakened the Fenton-like properties of the resultant materials, indicating that biogenic organic compounds play a significant role in Fenton-like reactions.

A further study was performed using EPR to clarify the active species generated by CSCM-SRB in the Fenton-like reaction. As Figure 8a shows, the signal peak at g = 2.004 represented that abundant S vacancy was found on CSCM-SRB. Also, signals of ·OH, ·O_2_^−^, and ^1^O_2_ were also found on the CSCM-SRB+H_2_O_2_ system, while no signals were found on the control H_2_O_2_ system (Figure 8b). Radical-trapping experiments were also performed and the results are shown in Figure 8c. It was found that ·O_2_^−^ and ·OH played significant roles in MB degradation, which corresponded with the EPR results.

Then, the generation of active species could be inferred. Firstly, with the electron from the S vacancy, O_2_ could be directly reduced to ^1^O_2_ (Equation (1)), which was able to degrade the organic pollutants. Secondly, during the Fenton-like reaction, Cu_2_S could react with H_2_O_2_ to generate the highly oxidizing ·OH, ^1^O_2_, and ·O_2_^−^ (Equations (2)–(6)), thus contributing to MB degradation [14,36,44,45,46].
2O_2_ + e^−^ (S vacancy) → ·O_2_^−^ → 2^1^O_2_(1)
Cu^+^ + H_2_O_2_ → Cu^2+^ + OH^−^ + ·OH(2)
Cu^2+^ + H_2_O_2_ → Cu^+^ + ·HO_2_ + H^+^(3)
Cu^2+^ + ·HO_2_ → Cu^+^ + ^1^O_2_ + H^+^(4)
·HO_2_ → H^+^ + ·O_2_^−^(5)
2·O_2_^−^ + 2H_2_O → ^1^O_2_ + H_2_O_2_ + 2OH^−^(6)

Based on the above results, a simple device for wastewater treatment was designed, as shown in Figure 9. Some H_2_O_2_ was added to untreated wastewater in advance and the pH and temperature were adjusted to make the reaction efficient. Then, the wastewater with added H_2_O_2_ slowly and stably passed through a filtration unit containing CSCM-SRB as the filtering membrane. After this filtration, the organic pollutants and bacteria were successfully removed by CSCM-SRB. It was found that the treated wastewater was a transparent and clear liquid. In this filtration unit, the Fenton-like mechanism of CSCM-SRB occurred. Through the Fenton-like reaction of Cu_2_S with H_2_O_2_, ^1^O_2,_ ·OH, and ·O_2_^−^ were generated. Especially with abundant S vacancy, the production of these active species was enhanced. With the high oxidizing properties of ^1^O_2,_ ·OH, and ·O_2_^−^, organic pollutants such as MB were degraded. At the same time, the active species also damaged microbial cell membranes and the DNA of pathogenic microorganisms, resulting in high disinfection effects. Hence, through this simple one-step treatment with the device we designed, organic pollutants and pathogenic microorganisms were efficiently removed, providing a novel method for one-step wastewater treatment.

## 4. Conclusions

In this paper, a novel method of Cu_2_S synthesis on copper mesh via electrolysis in SRB-cultured medium was proposed. The resultant CSCM-SRB showed good performance in terms of organic dye degradation and bacterial disinfection, revealing promising wastewater treatment prospects.

It was found that following electrolysis in SRB medium, squamous-like Cu_2_S arrays with a size of ~1 μm were obtained on the copper mesh, exhibiting typical orthorhombic characteristics. The loading contents of Cu_2_S varied with the electrolyzing parameters. CSCM-SRB with the highest catalytic MB degradation properties was obtained by electrolysis at 3.75 mA/cm^2^ for 900 s. The optimized reacting conditions were determined to be 1.2 cm^2^/mL CSCM-SRB with 0.05 M H_2_O_2_ at 35 °C when pH = 6. Under this condition, the degradation of MB reached over 99% after 120 min of reaction. Our novel material’s disinfection properties were also proven by antibacterial tests, revealing that an almost 100% antibacterial effect against *E. coli* was obtained after an 8 min reaction. The organic compounds produced by SRB and adsorbed on CSCM-SRB played important roles in the properties of CSCM-SRB. Finally, possible Fenton-like mechanisms of CSCM-SRB were proposed, illustrating that ·O_2_^−^, ·OH, and ^1^O_2_ acted as the main functional species during Fenton-like reactions, leading to effective MB degradation and high antibacterial properties. Based on these findings, a simple device with possible applications in wastewater treatment was designed.

## Figures and Tables

**Figure 1 nanomaterials-14-00621-f001:**
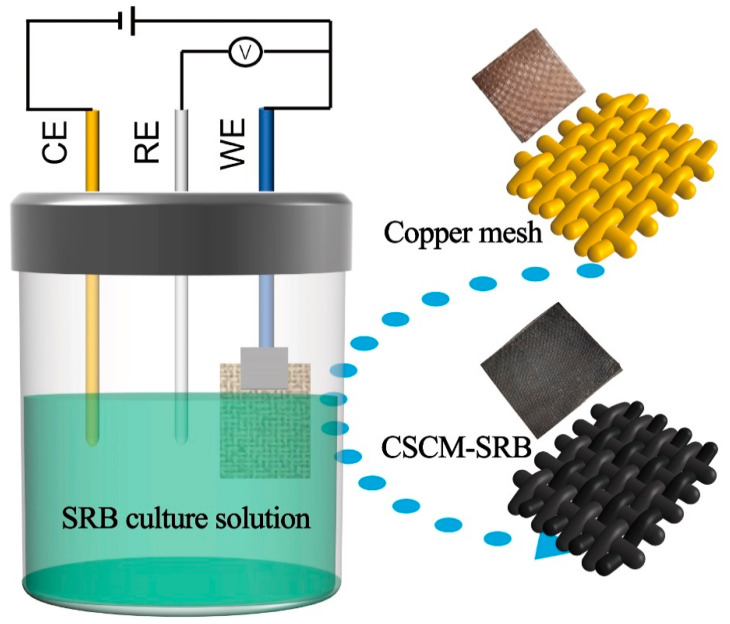
Synthesis of CSCM-SRB.

**Figure 2 nanomaterials-14-00621-f002:**
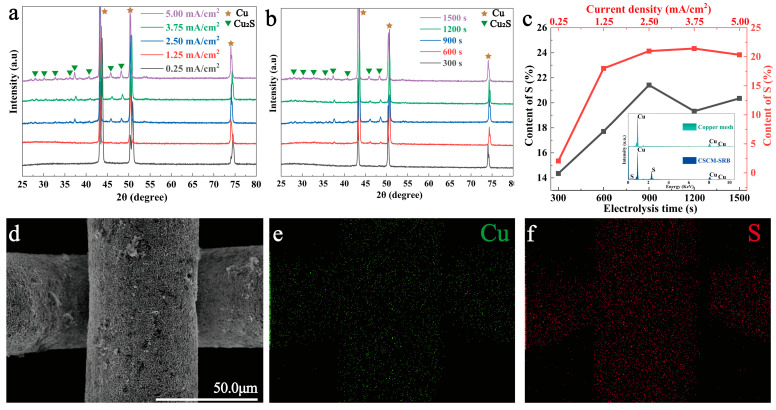
XRD patterns of CSCM-SRB obtained at different current densities (**a**) and different electrolysis times (**b**). Contents of S in CSCM-SRB obtained at different current densities (**c**) and different electrolysis times (**c**). EDS spectrum of copper mesh and CSCM-SRB (**d**) and EDS maps of CSCM-SRB (**e**,**f**).

**Figure 3 nanomaterials-14-00621-f003:**
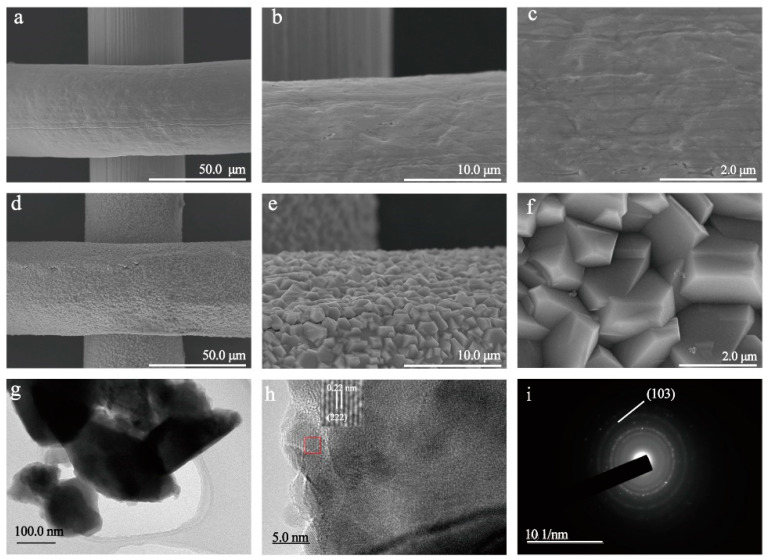
SEM images of copper mesh (**a**–**c**) and CSCM-SRB (**d**–**f**), and TEM images (**g**–**i**) of Cu_2_S synthesized on CSCM-SRB.

**Figure 4 nanomaterials-14-00621-f004:**
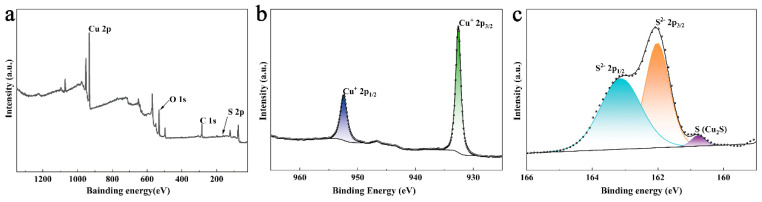
Survey spectrum (**a**) and high-resolution spectra of Cu 2p (**b**) and S 2p (**c**) in XPS spectra of CSCM-SRB.

**Figure 5 nanomaterials-14-00621-f005:**
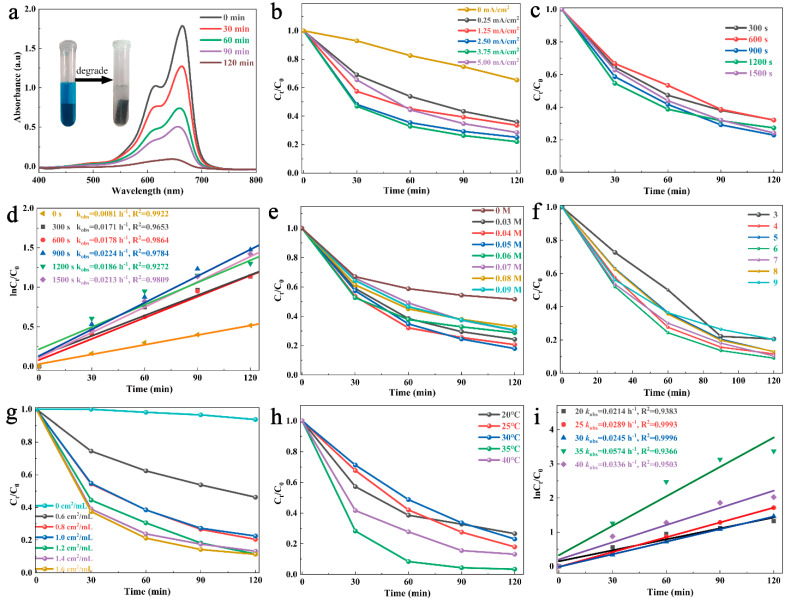
UV-Vis DRS spectrum of the MB system with CSCM-SRB (**a**), the degradation efficiencies of the system with CSCM-SRB electrolysis under various current densities (**b**) and various electrolysis times (**c**) with kinetic fitting analysis (**d**), and the degradation efficiencies of the CSCM-SRB system under an optimized H_2_O_2_ concentration (**e**), pH value (**f**), CSCM-SRB usage (**g**), and temperature (**h**) with kinetic fitting analysis (**i**).

**Figure 6 nanomaterials-14-00621-f006:**
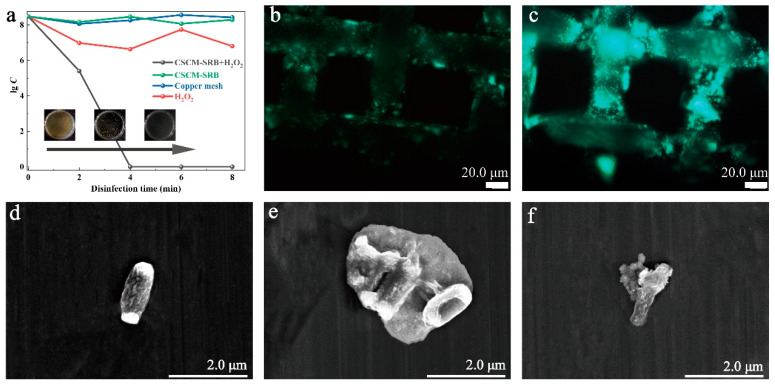
Logarithmic survival of bacterial concentration of *E. coli* against CSCM-SRB (**a**), fluorescence images of CSCM-SRB+H_2_O_2_ (**b**) and CSCM-SRB (**c**) after disinfection, and SEM images of untreated *E. coil* (**d**) and disinfected *E. coil* after 4 min (**e**) and 8 min (**f**) reaction.

**Figure 7 nanomaterials-14-00621-f007:**
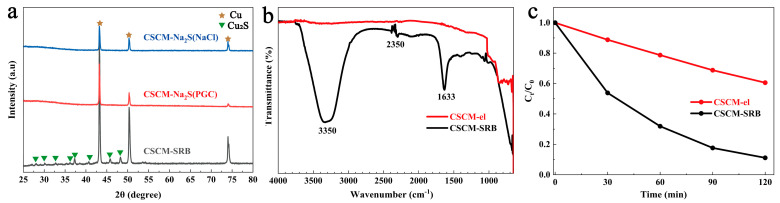
XRD patterns of CSCM electrolyzed in various electrolytes (**a**), FT-IR spectra of CSCM-SRB and CSCM-el (**b**), and the MB degradation efficiency of CSCM-SRB and CSCM-el (**c**).

**Figure 8 nanomaterials-14-00621-f008:**
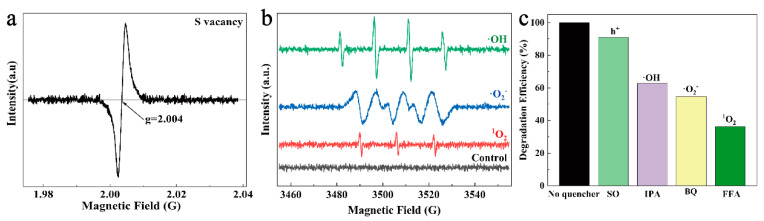
EPR signals of S vacancy (**a**), DMPO-OH, DMPO-O_2_^−^, TEMPO-^1^O_2_ (**b**) for CSCM-SRB, and radical-trapping results for MB degradation by CSCM-SRB (**c**).

**Figure 9 nanomaterials-14-00621-f009:**
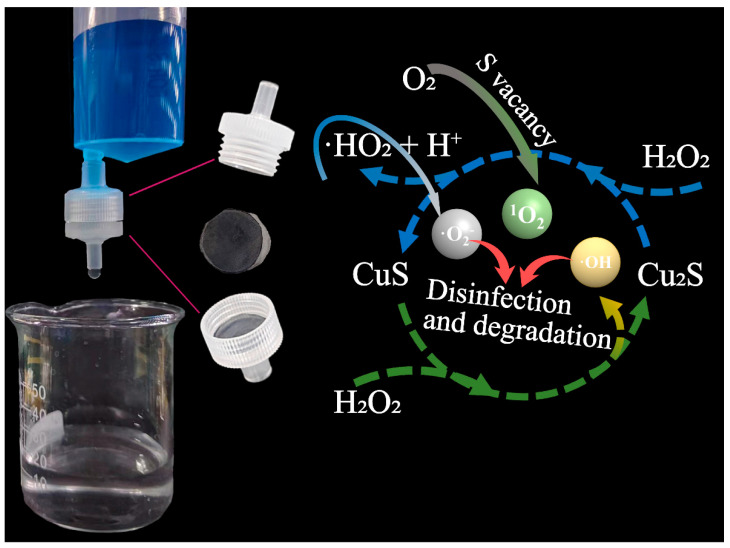
Design of novel device for wastewater treatment using CSCM-SRB with Fenton-like mechanism.

## Data Availability

Data are contained within the article and Appendix A.

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
