# Peer review of "In Situ Electrochemical Synthesis of Squamous-like Cu2S Induced by Sulfate-Reducing Bacteria as a Fenton-like Catalyst in Wastewater Treatment: Catalytic Performance and Mechanism"

_nanomaterials, 2024, doi:10.3390/nano14070621_

Round 1

Reviewer 1 Report

Comments and Suggestions for Authors

The manuscript presents a novel method for synthesis of Cu2S and covering a Cu mesh with that material (to obtain CSCM-SRB). The material is characterized by different methods and tested as a part of Fenton-like reagent for the degradation of methylene blue (MB). The effect of different parameters on the MB degradation have been studied. The synthesized material exhibited also disinfection ability with respect to Escherichia Coli. The clarification of the mechanism of antibacterial action can be pointed as a good achievement of the authors. Another positive feature of the work is the proposed possible Fenton-like mechanism of CSCM-SRB action in the MB degradation. However, in this case stoichiometry of the written equations have to be checked.

Clarification is needed regarding the order of the reaction model used.

I would not entirely agree with the conclusion that a simple device presented can be used for applications in real environments - pH, temperature, H2O2 doze have to be optimized. The most important obstacle, in my opinion, would be the needed ratio of 1.2 cm2 CSCM-SRB to one mL of water to be treated.

Some used terms have to be checked. Some corrections in the English language are necessary.

In order to facilitate authors, I am attaching the draft text file with comments and suggestions inside.

Comments on the Quality of English Language

Moderate editing of English language required

Reviewer 2 Report

Comments and Suggestions for Authors

This paper presents a method for the synthesis of copper sulfide (Cu2S) on copper mesh through electrolysis in a culture medium of sulfate-reducing bacteria (SRB). The authors describe their experimental approach and results, emphasizing the influence of electrolysis parameters on the formation of scale-like Cu2S arrays on copper meshes and the variance of Cu2S loading.

The resulting Cu2S-SRBs exhibit high catalytic properties for the degradation of methylene blue (MB), and the study also showed the antibacterial efficiency of almost 100% antibacterial rate against E. coli within an 8-min reaction time, indicating a strong disinfection effect of the developed system.

Overall, the manuscript is well written and well structured. However, some improvements should be considered to improve the overall quality of the text.

Figure 2 b: The current used in the study for the different electrolysis times is not indicated.

EDS results (Fig. 2 e to d) and EDS analysis: Details of the procedure for quantification with EDS are not given. There is no statistical analysis of these results. EDS is a semi-quantitative technique and special care must be taken when analysing these results. This analysis must be reported in detail.

Figure 5: Information is missing in the various diagrams in this figure because not all the specified conditions are explicitly stated. Further details on these experiments should be provided in the main text or in the caption to assist the reader.

The captions of figures 5h and 5f are ambiguous and do not seem to describe what is shown in these figures. A revision of these figures and captions is necessary.

The paper does not extensively address the reuse and regeneration of CSCM-SRB after treatment processes. Investigating the material's performance over multiple cycles and developing efficient regeneration methods are important.

Although the article suggests possible Fenton-like mechanisms, the specificity, and detailed pathways of these reactions, especially in complex real wastewater matrices, could be further elucidated. The proposed apparatus in Figure 9 is not very helpful in explaining this mechanism, and some things are asserted in this part without being substantiated in the literature. A revision is necessary.

Taking all of the above points into account, it is recommended that a major revision must be done before considering for publication.

Round 2

Reviewer 2 Report

Comments and Suggestions for Authors

My main concerns about this manuscript have been addressed by the authors. The recommendation is to accept it.